# Characteristics of Urine Organic Acid Metabolites in Nonalcoholic Fatty Liver Disease Assessed Using Magnetic Resonance Imaging with Elastography in Korean Adults

**DOI:** 10.3390/diagnostics12051199

**Published:** 2022-05-11

**Authors:** Ji-Hee Haam, Yun Kyong Lee, Eunkyung Suh, Young-Sang Kim

**Affiliations:** 1Chaum Life Center, CHA University, Seoul 06062, Korea; hamjhi@chamc.co.kr (J.-H.H.); ykleefm@chamc.co.kr (Y.K.L.); sherby@chamc.co.kr (E.S.); 2Department of Family Medicine, CHA Bundang Medical Center, CHA University, Seongnam 13496, Korea

**Keywords:** hepatic steatosis, hepatic fibrosis, nonalcoholic fatty liver disease, urine organic acid, metabolomics

## Abstract

The liver is an essential organ that manufactures energy through various metabolic pathways; thus, exploring the intermediate metabolites in nonalcoholic fatty liver disease (NAFLD) may help discover novel parameters in hepatic steatosis or fibrosis. The present study aimed to investigate the traits of urine organic acid metabolites in participants with hepatic steatosis and fibrosis in nonalcoholic Korean adults. Hepatic steatosis and fibrosis, in 68 men and 65 women, were evaluated using quantification by proton density fat fraction with magnetic resonance (MR) imaging and MR elastography, respectively. Urine metabolites were obtained using a high-performance liquid chromatography–mass spectrometry analysis. The candidate metabolites were included in the logistic regression models for hepatic steatosis and fibrosis. The association between high p-hydroxyphenyllactate levels and hepatic steatosis was not independent of body mass index and Homeostatic Model Assessment-insulin resistance. High ethylmalonate, β-hydroxybutyrate, and sulfate levels were significantly related to a low probability of hepatic fibrosis, independent of covariates. In conclusion, urine metabolites were not related to hepatic steatosis independent of obesity and insulin resistance, while several metabolites were specifically associated with hepatic fibrosis. Further study is required to verify the diagnostic value of the metabolites in a population with wide-spectrum NAFLD.

## 1. Introduction

Nonalcoholic fatty liver disease (NAFLD) is the most common cause of chronic liver disease worldwide [1] and is generally defined as hepatic fat accumulation without secondary causes [2]. NAFLD may progress to fibrosis, cirrhosis, and hepatocellular carcinoma [3,4,5,6]. Liver biopsy is the most reliable approach to identifying NAFLD [2,7]; however, non-invasive modalities are widely used [8]. Magnetic resonance imaging (MRI) distinguishes degrees of steatosis and fibrosis with good accuracy and sensitivity [9,10]. In particular, MR elastography (MRE) is regarded as the most important imaging technique for hepatic fibrosis [11].

Abnormal serum concentrations of transaminases, such as aspartate aminotransferase (AST) and alanine aminotransferase (ALT), indicate liver injury [12]. These tests are clinically utilized and well-validated; however, their values do not surrogate the status of NAFLD [13,14]. Hence, several equations, such as FIB-4 and the AST to platelet ratio index (APRI), and a series of biomarkers called the enhanced liver fibrosis (ELF) panel, are suggested to assess fibrosis in NAFLD [2,15,16]. Considering the liver as an essential organ that manufactures energy through various metabolic pathways [17], a liver abnormality may change the intermediate pathway products. The imbalance of liver energy metabolism, adipocyte dysfunction, and genetic causes are also related to NAFLD [18]. Several studies have investigated the metabolomic markers that are derived from the metabolism in NAFLD [19,20,21,22,23].

Urine organic acid tests have been used in the field of inborn errors of metabolic disorders [24]. The commercial kits for urine organic acid tests enable the detection of various molecules that are derived from endogenous and xenobiotic metabolism [25]. The characteristic investigation of intermediate metabolites may help to understand the common abnormalities in metabolism in hepatic steatosis or fibrosis. The present study aimed to investigate the traits of urine organic acid metabolites in participants with hepatic steatosis and fibrosis in nonalcoholic Korean adults. Additionally, it also evaluated whether the detected metabolites are independent of conventional indices associated with NAFLD.

## 2. Subjects and Methods

### 2.1. The Study Participants

This cross-sectional study was conducted based on the data from Chaum Life Center health checkups between November 2016 and January 2019. Initially, 145 participants underwent both urine organic acid metabolite analyses and liver MRI with elastography. Those with acute disease, thyroid disease, abnormal kidney function (estimated glomerular filtration rate of <60 mL/min/1.73 m^2^), and a history of stroke, angina, myocardial infarction, or any cancer were excluded. Participants with significant alcohol consumption (>21 and >14 standard drinks per week in men and women, respectively) [2], seropositivity in viral hepatitis (HBs Ag or HCV Ab), and long-term use of glucocorticoid were also excluded to rule out other causes of liver diseases. Finally, 68 men and 65 women were enrolled in this study. The study was conducted according to the guidelines of the Declaration of Helsinki and approved by the Institutional Review Board of CHA Bundang Medical Center (2018-07-026).

### 2.2. Measurements and Personal Medical History

Self-report questionnaires were used to collect information about participants’ medical history and lifestyle. Height and weight were measured in the standard position [26]. Blood pressure was measured using an automatic sphygmomanometer after resting for 10 min in a sitting position.

Blood samples were collected in the morning after the patient had fasted overnight for at least 8 h and drawn from the antecubital area. Serum samples were stored at 4 °C and analyzed within a day of sampling. Glucose, aminotransferases, creatinine, and lipid profiles were tested using an automatic analyzer (Hitachi 7600; Hitachi, Tokyo, Japan). Fasting plasma insulin concentrations were determined using an electrochemiluminescence immunoassay (Elecsys insulin, Roche, Mannheim, Germany). Insulin resistance (IR) was approximated using the Homeostatic Model Assessment (HOMA) calculator v2.2.3 (Oxford Center for Diabetes, Endocrinology and Metabolism, Oxford, UK; available at http://www.dtu.ox.ac.uk (accessed on 23 April 2022)). The FIB-4 index was calculated using the following formula, age [yr] × AST [U/L])/((PLT [10^9^/L]) × (ALT [U/L])^1/2^) [27].

### 2.3. Measurements of Urine Organic Acid Metabolites

Urine organic acid metabolites were measured using previously described methods and analyzed at Eone Laboratory, Inc. (Incheon, Korea) [26].

The concentration of each metabolite was normalized with urine creatinine level to minimize variability because of differences in urine concentration. The analyte levels were expressed using the unit of millimole per mole of creatinine.

### 2.4. Measurements of Hepatic Steatosis and Fibrosis

MRI was performed with a 1.5-T system (Signa HDxt, GE HealthcareMR, Milwaukee, WI, USA). Quantitatively encoded MRI was performed obtaining the proton density fat fraction (PDFF) maps of the liver. An MRI PDFF threshold of 6.4% diagnoses significant hepatic steatosis [28]. A pneumatic driver, which is a drum-like device that is designed to apply acoustic vibrations, was placed in the abdominal wall in a supine position and used to generate propagating mechanical waves in the liver. MRE uses propagating mechanical shear waves, which provide information for calculating elasticity. The shear elasticity (kPa) of the liver was measured as the mean value within the liver of the elasticity. A cutoff value of 2.5 kPa was used to distinguish patients with normal liver parenchyma from those with hepatic fibrosis [29,30].

### 2.5. Statistical Analysis

The general characteristics of the variables, including urine organic acid metabolites, were expressed as means ± SD, median (interquartile range), or number (proportion). Conventional indices, such as AST, ALT, gamma-glutamyl transferase (GGT), HOMA2-IR, and FIB-4, were compared between the participants with and without hepatic steatosis and fibrosis using the Mann—Whitney U test. The creatinine-adjusted values of the urine metabolite concentrations were categorized into quartiles, and the fourth quartile was considered as a high level. The high-level risks of each metabolite for hepatic steatosis and fibrosis were calculated using the chi-square test. Metabolites with *p*-values of <0.1 were included in logistic regression models. The odds ratios of high-level metabolites for hepatic steatosis and fibrosis that were adjusted for age and sex were estimated in Model 1. Liver parameter (ALT or FIB-4 for hepatic steatosis and fibrosis, respectively) was additionally adjusted to Model 1 in Model 2. Metabolic factors, such as body mass index and HOMA2-IR, were additionally adjusted (Model 3).

All statistical analyses were conducted using the SPSS statistical package, version 26 (IBM, Armonk, Westchester, NY, USA). Results with *p*-values of <0.05 were considered statistically significant.

## 3. Results

### 3.1. Characteristics of the Participants

The baseline characteristics of the participants are presented in Table 1. The mean age was 58.6 ± 10.9 years. Among the 133 participants, 68 (53.1%) were men. According to the cutoffs, 43.6% and 55.6% of the participants were classified into hepatic steatosis and fibrosis, respectively (Figure 1). Appendix A shows the characteristics of the urine organic acid metabolites.

### 3.2. Differences in Conventional Parameters According to Hepatic Steatosis and Fibrosis

Figure 2 displays the proportions of the participants according to hepatic steatosis and fibrosis. Additionally, Figure 3 shows the differences in conventional parameters according to hepatic steatosis and fibrosis. The levels of AST, ALT, GGT, and HOMA2-IR were significantly higher in participants with hepatic steatosis than those without, while AST and FIB-4 levels were higher in those with hepatic fibrosis than those with normal elasticity.

### 3.3. The Association of Hepatic Steatosis and Fibrosis with Urine Organic Acid Metabolites

Figure 4 and Appendix A show the ORs of high-level metabolites for hepatic steatosis and fibrosis. High succinate, vanillylmandelate, and pyroglutamate levels were significantly associated with lower hepatic steatosis prevalence. High formiminoglutamate and p-hydroxyphenyllactate levels were positively associated with hepatic steatosis. High sulfate levels were associated with lower hepatic fibrosis prevalence. Metabolites with a *p*-value of <0.1 were included in the logistic regression models (Table 2). Accordingly, pyruvate, homovanillate, and picolinate were additionally included in the models for hepatic steatosis, whereas ethylmalonate and β-hydroxybutyrate were included in the hepatic fibrosis models. High p-hydroxyphenyllactate levels were significantly associated with hepatic steatosis in the ALT-adjusted model (Model 2; OR = 3.379, 1.307–8.734), but not in the model that was further adjusted for metabolic factors (Model 3). High succinate levels had a trend to have low hepatic steatosis probability (*p* = 0.058). Other metabolites were not significantly related to hepatic steatosis independent of ALT. High sulfate levels were significantly associated with lower hepatic fibrosis probability than the other quartiles (Model 3; OR = 0.243, 0.097–0.610). Likewise, high ethylmalonate and β-hydroxybutyrate levels were significantly related to low hepatic fibrosis probability in fully adjusted models.

## 4. Discussion

The present study evaluated the characteristics of urine organic acid metabolites in NAFLD according to hepatic steatosis and fibrosis in Korean adults. Hepatic steatosis was associated with high p-hydroxyphenyllactate levels; however, the association was not independent of metabolic factors. Hepatic fibrosis was inversely associated with high ethylmalonate, β-hydroxybutyrate, and sulfate levels.

Expectedly, our study revealed a significant association between hepatic steatosis and liver enzymes and IR, which may indicate that hepatic steatosis and elevated liver enzymes are strongly linked with systemic obesity and metabolic syndrome. Contrastingly, hepatic fibrosis was associated with calculated FIB-4, but not with serum ALT and GGT levels. The discordance between hepatic fibrosis and serum liver enzyme levels led to the need for a better hepatic fibrosis index. Therefore, various equations, such as FIB-4, APRI, and NAFLD fibrosis score, were developed and utilized [2,15,16,31,32]. Some biologic markers other than the conventional liver enzymes were introduced into the ELF panel to detect advanced fibrosis [33,34]. Metabolomics has been investigated in NAFLD and nonalcoholic steatohepatitis, which provides new perspectives in diagnosing and identifying novel biomarkers [19,20,21,22,23].

NAFLD has been known to accompany significant changes in the hepatocyte, such as enhanced gluconeogenesis, lactate production, and tricarboxylic acid (TCA) cycle, and decreased ketone body production, mitochondrial respiration, and adenosine triphosphate synthesis [35]. Our study revealed that β-hydroxybutyrate and ethylmalonate were inversely associated with hepatic fibrosis. β-Hydroxybutyrate is the most abundant ketone in the human body [36], and ethylmalonate is generally formed from butyrate [37]. Ethylmalonic aciduria is known to be associated with elevated butyryl-CoA levels from isoleucine catabolism [38,39]. Considering that both β-hydroxybutyrate and ethylmalonate are derived from fatty acid breakdown [40], fatty acid metabolism abnormality may be an indicator of hepatic fibrosis.

The ratio of urinary sulfate to creatinine indicates the total body reserve of sulfur-containing compounds, including glutathione, which is used in phase II pathways [41,42]. Phase II drug-metabolizing enzymes play an important role in the biotransformation of endogenous compounds and xenobiotics to more easily excretable forms, as well as in the metabolic inactivation of pharmacologically active compounds [43]. Phase II enzyme dysfunctions may potentiate oxidative stress and liver injury [44].

Among the candidate metabolites for the association with hepatic steatosis, p-hydroxyphenyllactate was significant before the metabolic factor adjustment. This metabolite is derived from tyrosine catabolism by the gut microbiota [45] and was associated with obesity markers [46]. A study on the gut microbiome has shown that a link between the gut microbiome and p-hydroxyphenyllactate shares a gene effect with hepatic steatosis and fibrosis [47]. Moreover, urine p-hydroxyphenyllactate indicated hepatic encephalopathy in patients with hepatic cirrhosis [48]. These findings are consistent with our results. Another metabolite, succinate, had a trend of inverse association with hepatic steatosis. Succinate, a TCA cycle intermediate, has an antilipolytic effect in adipocytes [49]. Mitochondrial dicarboxylate carrier (mDIC), which is dominantly expressed in the white adipose tissue, is responsible for controlling the release of free fatty acids from adipocytes to the liver through the export of succinate from the mitochondria. Eventually, adipose mDIC-mediated succinate transport out of the mitochondrial matrix impacts adipocyte lipolysis and liver lipid accumulation [49].

Our study revealed no relationship between hepatic steatosis and fibrosis. Similarly, previous studies have shown that hepatic fibrosis was not associated with steatosis or steatohepatitis [50,51]. Additionally, our study revealed that metabolites associated with hepatic fibrosis and steatosis do not overlap each other. Moreover, most of the candidate metabolites, such as formiminoglutamate, vanillylmandelate, and picolinate, were not independent indices of hepatic steatosis, but markers linked to metabolic syndrome [26]. Considering that the associated metabolites with hepatic fibrosis were not influenced by metabolic parameters of obesity and IR as well as FIB-4, the deficiency of these metabolites in urine may be significant for early changes in hepatic fibrosis of NAFLD.

Our study has several limitations. First, the causal relationship between NAFLD and the intermediate metabolites was not confirmed because the study employed a cross-sectional design. The experimental design may be required to be aware of the changes in metabolites due to NAFLD changes or vice versa. Second, the definition of hepatic steatosis and fibrosis was not based on histologic findings. Additionally, the cutoff values of liver fat content and elasticity measured using MRI with elastography are different according to various studies [28,29,52,53,54,55]. Considering that our study participants were enrolled from health checkups and are relatively healthy, the cutoff value for elasticity was adopted at a low level [29,30]. The grades of hepatic steatosis and fibrosis need to be further categorized in a population, including more severe cases. Nonetheless, MRI with elastography is the best imaging tool for NAFLD at present [56,57]. Third, only the urine sample was analyzed for the intermediate metabolites. Additionally, diet type was not restricted, and food diary was not surveyed. However, the participants were exposed to a typical Korean diet and forbidden from special dietary materials, such as alcohol, coffee, fruits, and supplements, for a day. Further survey on diet habits may help interpret the association between NAFLD and urine metabolites. Overnight fasting was also mandatory for the participants. Further analyses may be required for more kinds of metabolites and from various specimens, including blood and liver biopsy.

## 5. Conclusions

Urine metabolites were not related to hepatic steatosis independent of obesity and IR, whereas several metabolites were specifically associated with hepatic fibrosis in Korean adults without significant alcohol consumption. Abnormalities in fatty acid metabolism and phase II detoxification may be important findings for early fibrotic changes in NAFLD. Further study is required to verify the diagnostic value of the metabolites in a population with a wide NAFLD spectrum.

## Figures and Tables

**Figure 1 diagnostics-12-01199-f001:**
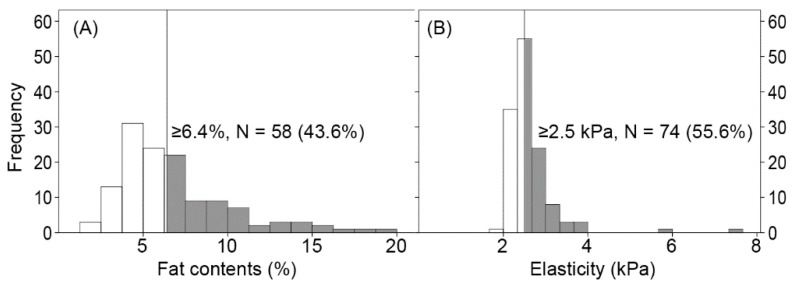
The histograms for liver fat content (**A**) and shear elasticity (**B**). Hepatic steatosis and fibrosis are defined according to the cutoffs of fat content and elasticity, respectively.

**Figure 2 diagnostics-12-01199-f002:**
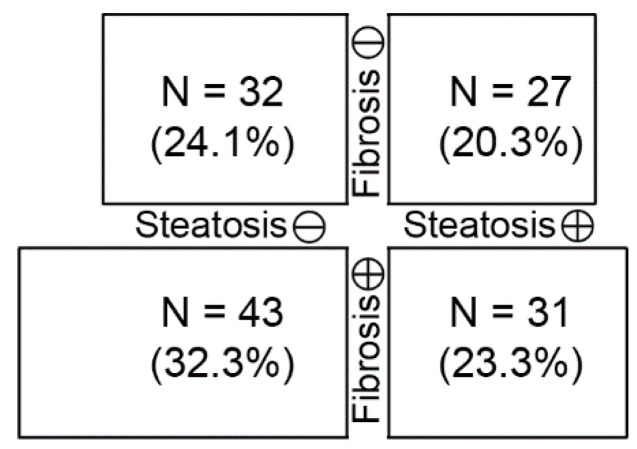
The participant numbers in each subgroup according to hepatic steatosis and fibrosis. Box size indicates the participant number.

**Figure 3 diagnostics-12-01199-f003:**
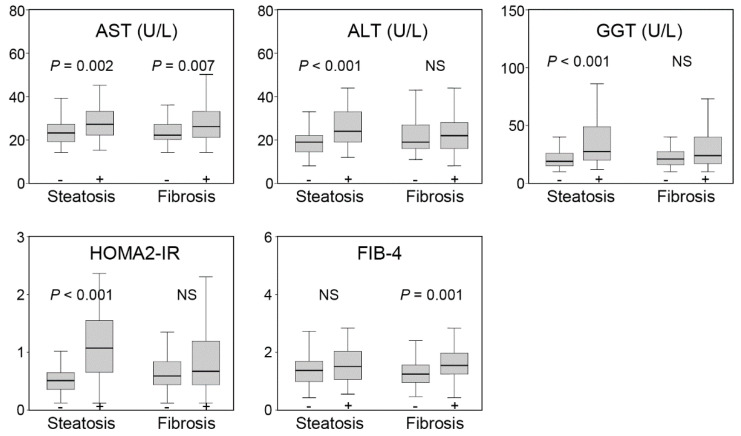
The liver parameter levels according to hepatic steatosis and fibrosis. Boxplots show the median, interquartile range, and minimum/maximum (except for outliers) of each parameter. *p* values were calculated using Mann—Whitney U tests.

**Figure 4 diagnostics-12-01199-f004:**
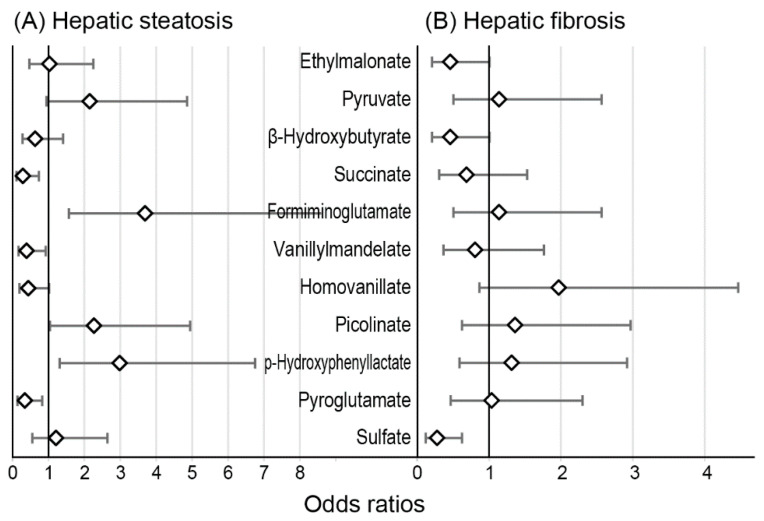
The odds ratios of high-level metabolites for hepatic steatosis (**A**) and fibrosis (**B**). Error bars show 95% CI.

**Table 1 diagnostics-12-01199-t001:** General characteristics of the study participants.

*n* = 133	
Age (years)	58.6 ± 10.9
Sex (men)	68 (51.1%)
**Medication history**	
Hypertension	31 (23.3%)
Diabetes	42 (31.6%)
Dyslipidemia	13 (9.8%)
**Anthropometry and measurements**	
Body mass index (kg/m^2^)	23.2 (21.2–25.4)
Systolic BP (mmHg)	119 (108–129)
Diastolic BP (mmHg)	74 (66–80)
**Laboratory results**	
AST (U/L)	24 (20–29)
ALT (U/L)	21 (16–28)
GGT (U/L)	22 (16–33)
HOMA2-IR	0.63 (0.44–1.11)
FIB-4	1.42 (1.04–1.84)
Urine creatinine (mmol/L)	12.2 (8.1–15.7)
**MRI with elastography findings**	
Fat content (%)	5.8 (4.3–8.2)
Elasticity (kPa)	2.51 (2.31–2.74)

Data are expressed as mean ± SD, median (interquartile range), or number (proportion). BP, blood pressure; AST, aspartate aminotransferase; ALT, alanine aminotransferase; GGT, gamma-glutamyl transferase; HOMA2-IR, homeostatic model assessment of insulin resistance; MRI, magnetic resonance imaging.

**Table 2 diagnostics-12-01199-t002:** The logistic regression models of the metabolites for hepatic steatosis and fibrosis.

	Model 1 (Age and Sex)		Model 2 (Liver Parameter)		Model 3 (Metabolic Factors)	
	OR	*p*	OR	*p*	OR	*p*
**Hepatic steatosis**						
Pyruvate	2.318 (0.934–5.756)	0.070	2.061 (0.781–5.437)	0.144	1.265 (0.415–3.856)	0.679
Succinate	0.371 (0.137–1.007)	0.052	0.410 (0.149–1.133)	0.086	0.327 (0.103–1.039)	0.058
Formiminoglutamate	2.174 (0.862–5.480)	0.100	1.250 (0.450–3.476)	0.669	0.797 (0.245–2.592)	0.706
Vanillylmandelate	1.003 (0.345–2.917)	0.995	0.589 (0.180–1.931)	0.382	0.800 (0.223–2.864)	0.732
Homovanillate	0.838 (0.296–2.376)	0.740	1.120 (0.373–3.359)	0.840	0.970 (0.274–3.431)	0.962
Picolinate	2.081 (0.875–4.949)	0.097	1.865 (0.755–4.604)	0.177	1.347 (0.480–3.781)	0.571
p-Hydroxyphenyllactate	3.078 (1.246–7.604)	0.015	3.379 (1.307–8.734)	0.012	2.665 (0.934–7.604)	0.067
Pyroglutamate	0.617 (0.225–1.693)	0.349	0.499 (0.168–1.483)	0.211	0.750 (0.226–2.482)	0.637
**Hepatic fibrosis**						
Ethylmalonate	0.444 (0.195–1.009)	0.053	0.385 (0.160–0.924)	0.033	0.390 (0.162–0.939)	0.036
β-Hydroxybutyrate	0.504 (0.225–1.130)	0.096	0.391 (0.163–0.937)	0.035	0.393 (0.162–0.956)	0.039
Sulfate	0.246 (0.105–0.580)	0.001	0.233 (0.094–0.580)	0.002	0.243 (0.097–0.610)	0.003

Model 1 was adjusted for age and sex, Model 2 was additionally adjusted for liver parameters (alanine aminotransferase for hepatic steatosis and FIB-4 for hepatic fibrosis, respectively), and Model 3 was additionally adjusted for BMI and HOMA2-IR.

## Data Availability

No new data were created or analyzed in this study. Data sharing is not applicable to this article.

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
