# Peer review of "Characteristics of Urine Organic Acid Metabolites in Nonalcoholic Fatty Liver Disease Assessed Using Magnetic Resonance Imaging with Elastography in Korean Adults"

_diagnostics, 2022, doi:10.3390/diagnostics12051199_

Round 1

Reviewer 1 Report

The article "characteristics of urine organic acid metabolites in ninakcoholic fatty liver disease assesed using magnetic resonance imaging with elastography in Korean adults" suggest that some urine organic test are related to hepatic fibrosis. The manuscript is clear and well written. My main concern  is the lack of histological studies so fibrosis and steatosis diagnosis was performed by MRI techniques. However, authors explained this concern as a study limitation. The analysis of dietary habits such as alcohol or fatty aliments may be also interesting because these are factors that are very related to fibrosis. Authors have included this concern as a limitation but a deeper discussion about diet habits, hepatic fibrosis and these urine metabolites would be interesting.

Author Response

Response: Thank you for the reviewer’s comment. As the reviewer commented, we know the limitation of the hepatic steatosis and fibrosis definition. In addition, dietary change may influence the association between NAFLD and urine metabolites. We added a sentence into the limitation section to emphasize the diet habit as follows: Further survey on the diet habits may help interpret the association between NAFLD and urine metabolites.

Reviewer 2 Report

Metabolite concentrations were categorized into quartiles, and the fourth quartile was considered as a high level. But could be better to calculate the high risk cut point with ROC Analysis instead of arbitrary fourth quartile.

In Figure 1, cutoffs of fat contents and elasticity should be clarified the criterion for it and whether it is applicable to each age and sex.

In Figure 1, The Power of the test should have been indicated considering the broad standard deviations in those who were positively diagnosed with Fibrosis and Steatosis particularly in the HOMA2-IR, GGT indices. The cut-off point may be inaccurate and if it had been examined in ROC analysis the standard deviation might have been smaller.

Author Response

Metabolite concentrations were categorized into quartiles, and the fourth quartile was considered as a high level. But could be better to calculate the high risk cut point with ROC Analysis instead of arbitrary fourth quartile.

Response: Thank you for the kind comments. We agree with the reviewer’s comment. ROC curve is one of good methods to determine the diagnostic cutoffs. We tried to determine the cutoffs of the metabolites for hepatic fibrosis using ROC curve and Youden index. High-level ethylmalonate, β-hydroxybutyrate, and sulfate were 35.3%, 27.8%, and 25.6%, respectively. The proportions were slightly higher than or similar to the proportions of the 4th quartiles. Additionally, the odds ratios were 0.382 (0.184–0.794; P=0.015), 0.428 (0.197–0.928; P=0.048), and 0.273 (0.120–0.625; P=0.003) in each metabolite, respectively. These risk ratios are also consistent with the results that we have submitted. We are handling 46 kinds of metabolites. In the next study, we hope that we may focus a few candidate metabolites and determine the detailed cutoffs.

In Figure 1, cutoffs of fat contents and elasticity should be clarified the criterion for it and whether it is applicable to each age and sex.

Response: As the reviewer criticized, the criteria for hepatic steatosis and fibrosis varied according to the articles. We described the evidence to determine the cutoffs for hepatic steatosis and fibrosis using MRI PDFF and elastography in the Section 2.4 measurements of hepatic steatosis and fibrosis. We also added the limitation of the definition of hepatic steatosis and fibrosis in the limitation part of Discussion.

In Figure 1, The Power of the test should have been indicated considering the broad standard deviations in those who were positively diagnosed with Fibrosis and Steatosis particularly in the HOMA2-IR, GGT indices. The cut-off point may be inaccurate and if it had been examined in ROC analysis the standard deviation might have been smaller.

Response: The power is an important issue to implement a study. We also agree that HOMA2-IR and GGT have broad SDs. In this reason, we used non-parametric methods to compare the indices like HOMA2-IR and GGT between the two groups. Nonetheless, we tried to calculate the power to compare the HOMA2-IR means between the groups with and without hepatic steatosis using the formula (https://en.wikipedia.org/wiki/Power_of_a_test). Fortunately, the power was 100%.

As the reviewer criticized, the cutoffs may be inaccurate. However, we applied some literatures to define the hepatic steatosis and fibrosis. To analyze the cutoffs for hepatic steatosis and fibrosis using ROC curves, we need the golden standard for diagnosis like biopsy results. In this aspect, our study has a limitation, and it has been described already.